# Common Causes for Sudden Shifts: Linking Phase Transitions in Sinusoidal Networks

## Abstract

Different phases of learning dynamics exist when training deep neural networks. These can be characterised by statistics called order parameters. In this work we identify a shared, underlying mechanism connecting three seemingly distinct phase transitions in the training of a class of deep regression models, specifically Implicit Neural Representations (INRs) of image data. These transitions include: the emergence of wave patterns in residuals (a novel observation), the transition from fast to slow learning, and Neural Tangent Kernel (NTK) alignment. We relate the order parameters for each phenomenon to a common set of variables derived from a local approximation of the structure of the NTK. Furthermore, we present experimental evidence demonstrating these transitions coincide. Our results enable new insights on the inductive biases of sinusoidal INRs.

## 1 Introduction

An increasing body of research emphasizes the pivotal role played by the optimization algorithm in learning representations. Indeed, it is known that the full expressivity of Deep Neural Networks (DNNs) is constrained in practice by the limitations of Gradient Descent (GD) in exploring the loss landscape (Hanin & Rolnick, 2019; Nye & Saxe, 2018). Furthermore, it has been established that neural networks learn patterns of different complexity at different rates, resulting in distinct learning phases (Arpit et al., 2017; Zhang & Wu, 2020). These phases may be identified through the collective evolution of model weights, which are quantified by summary statistics known as order parameters (Feng & Tu, 2021a; Stephenson & Lee, 2021; Liu et al., 2022; Ziyin & Ueda, 2022). In statistical mechanics, these parameters quantify symmetries of a system, and change suddenly at a phase transition (Sethna, 2021). Although various statistics have been independently identified in the DNN literature (Shwartz-Ziv & Tishby, 2017; Fort et al., 2020b; Sankararaman et al., 2020; Feng & Tu, 2021b), no underlying symmetry connects them, and their interrelationships remain unclear. Moreover, while order parameters can indicate the timing of phase transitions, they offer limited insight into what DNNs learn during these phases.

Part of the difficulty is that GD dynamics are nonlinear, and take place in a high-dimensional parameter space, where the model's relationship with the training data is obscured. The Neural Tangent Kernel (Jacot et al., 2018) (NTK) provides a complementary perspective that manifestly describes how local errors influence one another during training. Critically, in a phenomenon known as Neural Tangent Kernel Alignment (NTKA), the NTK undergoes significant changes early in training as it engages in feature learning, aligning with the target function. NTKA has been widely documented and is suggested as a reason why real-world DNNs often outperform their infinite-width limit counterparts (Hanin & Nica, 2019; Huang & Yau, 2019; Aitchison, 2020; Chizat et al., 2020; Lee et al., 2020; Seleznova & Kutyniok, 2022). Despite repeated empirical demonstrations of NTKA, theoretical exploration of the phenomenon has been largely restricted to simplified models and classification problems, leaving a gap in understanding the transition in complex regression tasks.

In this work, we extend the analysis of NTKA to Implicit Neural Representations (INRs), and in particular, SIREN models (Sitzmann et al., 2020). INRs are DNNs that model signals with low-dimensional domains, such as images, and are increasingly used as an alternative to discretized signal representations. Several works have studied static INR NTKs (Tancik et al., 2020; Yüce et al., 2022; Liu et al., 2023; Saratchandran et al., 2024), but this is known to offer a poor approximation (Vonderfecht & Liu, 2024). Our study is structured around four primary contributions:

1. We derive novel approximations for the local structure of the SIREN NTK, and use these structural parameters to derive formula for: the principal eigenvector (3.3); order parameters such as the minimum value of the Cosine NTK (3.4); and the correlation lengthscale (3.2). In so doing, we theoretically establish connections between the onset of NTK alignment and the location of inflection points in the training curve.

2. We identify a novel learning phase, characterized by the appearance of diffusion-like wavecrests in the residuals, and relate this behaviour to the NTK's evolution at this stage in the learning process.

3. We propose a new order parameter, MAG-MA (3.5), to capture a unifying mechanism underlying the different phase transitions. Motivated by the dependence of the order parameter approximations on spatial variations in the gradient, MAG-MA explicitly quantifies translational symmetry breaking in the local structure of the NTK.

4. We experimentally verify that the critical points for these different phase transitions cluster in time. We also empirically investigate the impact of image complexity and SIREN hyperparameters on the occurrence and timing of phase transitions and provide evidence that suggests NTK alignment in SIREN (image regression tasks) occurs in response to difficulties in modelling edges.

## 2 PRELIMINARIES

In this work, we consider the class of INRs that model 2D grayscale images, where pixel coordinates and their intensity form a dataset $\mathcal{D}$ of $N$ samples indexed with $i$, $(x_i, I(x_i))$, where $x_i \in \mathbb{R}^2$ and $I : \mathbb{R}^2 \mapsto \mathbb{R}$. On this dataset, we fit SIREN models $f$ with parameters $\theta$, using sinusoidal activation functions. In the continuum limit, we assume the data is distributed uniformly according to $P_{data}(x) = \text{Vol}(\mathcal{D})^{-1}$. We identify two fields: the local residual field $r(x; \theta(t)) = I(x) - f(x; \theta(t))$, and gradient field $\nabla_\theta f(x; \theta(t))$. Their time evolution is induced by gradient flow $\dot{\theta} = -\nabla_\theta L$ on the mean square error: $L(\theta) = \frac{1}{2\text{Vol}(\mathcal{D})} \int dx \; r(x; \theta)^2$. Accordingly, via the chain rule, the residuals evolve as follows:

$$\dot{r}(x; \theta(t)) = \nabla_\theta r(x; \theta(t)) \cdot \dot{\theta} \tag{1}$$

$$= -\frac{1}{\text{Vol}(\mathcal{D})} \int dx' \; r(x') \nabla_\theta r(x; \theta(t)) \cdot \nabla_\theta r(x'; \theta(t)) \tag{2}$$

$$= -\int dx' \; r(x') \underbrace{\left( \frac{1}{\text{Vol}(\mathcal{D})} \nabla_\theta f(x; \theta(t)) \cdot \nabla_\theta f(x'; \theta(t)) \right)}_{K_{NTK}(x, x'; \theta(t))} \tag{3}$$

In the last line, we defined the NTK. For notational brevity, we will drop the explicit dependence on $\theta$, and write $x' = x + u$. We also define a kernel closely related to the NTK, the Cos NTK:

$$C_{NTK}(x, x + u) = \frac{1}{\text{Vol}(\mathcal{D})} \frac{\nabla_\theta f(x) \cdot \nabla_\theta f(x + u)}{||\nabla_\theta f(x)|| \, ||\nabla_\theta f(x + u)||} \tag{4}$$

## 3 DERIVING ORDER PARAMETERS FROM THE NTK

We illustrate the different phases of learning with a motivating example. In Figure 1, we train a five-layer deep, 256-unit wide SIREN model on a $128 \times 128$ grayscale image of a camera-man, using full-batch GD with a learning rate of $10^{-3}$ for 2000 epochs. Our validation task is super-resolution reconstruction of the original image. During training, we examine the model's behaviour through three different lenses, with a sudden, qualitative shift revealed in each.

While these shifts are visually striking, in this section, we take a more quantitative approach based on the identification of order parameters. We then demonstrate why these phase transitions occur simultaneously, by relating the order parameters to a common set of features, which control the local structure of the NTK. The three lenses are as follows:

- **Spatial Distribution of Residuals**: Early in training, the loss decreases uniformly over the dataset (Drift Phase). However, at a critical point, we observe the formation of "wave-crests" corresponding to regions of low-loss, which propagate across the dataset (Diffusion Phase). To the best of our

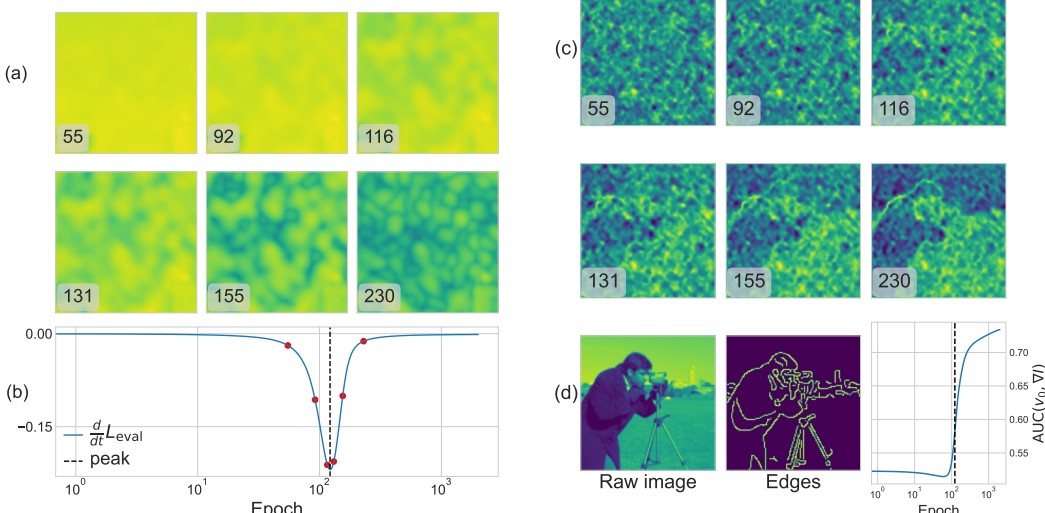

Figure 1: **A Single Phase Transition Through Three Lenses**: (a) The spatio-temporal evolution of the loss, as revealed through the magnitude of the residuals. Near the critical point we see the formation of wavecrests. (b) Evolution of the loss rate during training. The rate of change of the validation loss reaches a peak at the critical point. (c) Evolution of the principle eigenvector of the NTK reveals a sudden shift from disorder to learned features. (d) Quantification of NTKA in terms of alignment between edges and the principal eigenvector.

knowledge, we are the first to report this behaviour in SIREN models. In Section 3.1, we attribute this behaviour to changes in the equal-time correlation functions of the gradient field $\nabla_\theta f(x)$, whose parameters we derive in Section 3.2.

- **Principal Eigenvectors of the NTK**: Early in the training, the principal eigenvector $v_0$ is static and appears as a highly-disordered, structureless image (Disordered Phase). However, at a critical point, $v_0$ rapidly aligns with the edges of the image (Aligned Phase), after which it becomes static again. Though NTK alignment has been previously studied in the context of classification problems (Kopitkov & Indelman, 2020; Baratin et al., 2021; Shan & Bordelon, 2022; Canatar & Pehlevan), there are additional subtleties to consider for a regression task like INR training. To this end, we introduce a metric, $\text{AUC}(v_0, \nabla I)$ in Section 3.3 to identify when alignment occurs. We also derive an approximation of $v_0$ based on the local structure of the NTK, as outlined in Sections 3.1 and 3.2.

- **Training Curve Analysis**: There is a rapid shift in the slope of the training curve, which we call the loss rate $\dot{L}$. Initially, $\dot{L}$ is large, indicating the Fast Phase. After a critical point, the loss rate collapses, and learning slows (Slow Phase). Several works have studied this transition using order parameters, but in this work, we focus on the concept of gradient confusion, as described in (Fort et al., 2020b), (Sankararaman et al., 2020), (Feng & Tu, 2021b). In Section 3.4, we derive an approximation of this parameter based on the local structure of the NTK outlined in Section 3.2.

Having united the different order parameters, we are in a better position to speculate on the common origin of the underlying phase transition. Motivated by the dependence of each parameter on spatial variations in the magnitude field $||\nabla_\theta f(x)||$, we introduce a new parameter, termed MAG-Ma, in Section 3.5. MAG-Ma explicitly tracks violations in the translational symmetry of the NTK.

### 3.1 CORRELATION FUNCTIONS AND THE ONSET OF DIFFUSION

The form of equation 3 is reminiscent of the linear response functions in statistical field theory (Sethna, 2021; Goodman, 1985): to find the rate of change of the residual field at a point $x$, the kernel $K$ aggregates information about the residual at points $x + u$. To quantify the range of these interactions, we examine the local, equal-time correlation functions for the gradients $\nabla_\theta f(x)$ separated by a

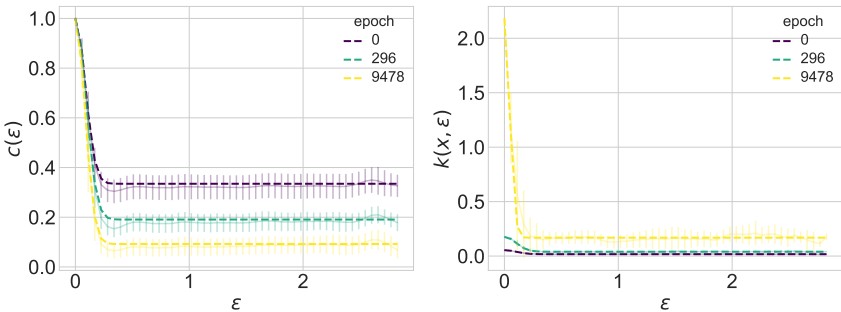

Figure 2: Visualization showing the empirical correlation function for the normalized parameter gradients. On the left-hand side is the global correlation-function for the $C_{NTK}$. On the right is the local-correlation function for the $K_{NTK}$ around a test point $x$. Dashed lines show fitted Gaussian approximation, and error bars show variance across dataset. Over the course of training, both the global correlation lengthscale $\xi_{corr}$, and the terminal value $c_\infty$, evolve.

distance $\epsilon$:

$$k(x, \epsilon) = \mathbb{E}_\phi\big[\nabla_\theta f(x) \cdot \nabla_\theta f(x + \epsilon \hat{e}_\phi)\big] \tag{5}$$

$$= \mathbb{E}_\phi\big[K_{NTK}(x, x + \epsilon \hat{e}_\phi)\big] \tag{6}$$

Here, $\hat{e}_\phi$ denotes a unit vector in direction $\phi$. Similarly, the global, equal-time correlation is given by:

$$k(\epsilon) = \mathbb{E}_x\big[k(x, \epsilon)\big] \tag{7}$$

Here, the expectation is taken uniformly over the unit vectors $\hat{u}$. We may define similar quantities for the $C_{NTk}$, which we denote by $c(x, \epsilon)$ and $c(\epsilon)$. We expect the range of these interactions to be short, as INRs are often carefully designed to ensure a diagonally dominant NTK(Tancik et al., 2020; de Avila Belbute-Peres & Kolter, 2022; Liu et al., 2023). To verify this, we group pairs of datapoints based on their distance, and then compute the mean $C_{NTk}$ value. For SIREN models, we observe that the equal-time correlation functions are well-approximated by Gaussians of the form[1]:

$$c(\epsilon) \approx (1 - c_\infty)e^{-\epsilon^2/2\xi_{corr}^2} + c_\infty \tag{8}$$

$$k(x, \epsilon) \approx ||\nabla_\theta f(x)||^2 (1 - c_\infty(x))e^{-\epsilon^2/2\xi(x)^2} + ||\nabla_\theta f(x)||^2 c_\infty(x) \tag{9}$$

This approximation introduces two important order parameters: the first, the correlation length-scale $\xi_{corr}$, controls the rate at which correlations decay with distance, defining the range of interactions. The second, the asymptotic value $c_\infty$, describes the interactions between points at separations $\epsilon$ much greater than $\xi$, where the gradient field vectors become uncorrelated. We have:

$$\lim_{\epsilon \to \infty} c(\epsilon) = c_\infty = \left|\left|\mathbb{E}_x\left[\frac{\nabla_\theta f(x)}{||\nabla_\theta f(x)||}\right]\right|\right|^2 \tag{10}$$

Dynamically, we see from Figure 2 that both $\xi$ and $c_\infty$ evolve during training, and we shall demonstrate that changes in these values account for the onset of diffusion. When $c_\infty$ decays to zero, we have, as a very simple approximation of the NTK:

$$K(x, x + u) \approx ||\nabla_\theta f(x)||^2 \exp(-||u||^2/\xi^2(x)) \tag{11}$$

When $\xi(x)$ is small, the NTK will suppress all contributions to the residual $\dot{r}(x)$ except from the immediate vicinity of $x$. As such, performing a Taylor expansion to second order in $u$, we obtain:

$$r(x + u; \theta) \approx r(x; \theta) + u^\top \nabla_x r(x; \theta) + \frac{1}{2}u^\top \nabla_x^2 r(x; \theta)u \tag{12}$$

Inserting this, along with the NTK approximation, into equation 3, the full integral may be solved using Gaussian integration (full details in Appendix A.2). We obtain:

$$\frac{d}{dt}r(x; \theta) = -2\pi\xi^2(x)||\nabla_\theta f(x)||^2 r(x) - \pi\xi^4(x)||\nabla_\theta f(x)||^2 \Delta^2 r, \tag{13}$$

which resembles a standard diffusion equation.

---

[1]We examine this assumption qualitatively in Figure 13, and numerically in Appendix F.1.

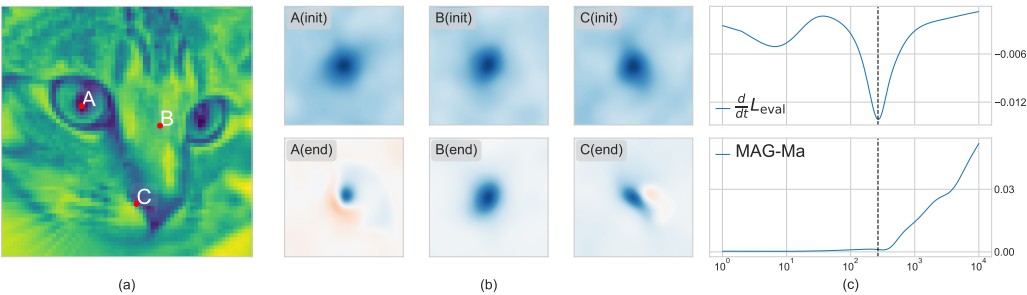

Figure 3: **Evolution of the Cosine NTK**: We visualize $C_{NTK}(x, x + u)$ around three points $x \in \{A, B, C\}$ for small separations $u$. At initialization, $C_{NTK}$ locally resembles an isotropic, translation-invariant RBF. However, as training progresses, these symmetries are broken. MAG-Ma (described in Section 3.5) is an order-parameter that monitors the original symmetry, and changes at the critical point.

## 3.2 BEYOND THE ISOTROPIC GAUSSIAN APPROXIMATION

Though the isotropic Gaussian approximation of the NTK can explain the appearance of the diffusion wavecrests, empirically, the NTK is anisotropic (see Figure 3). What's more, the isotropic Gaussian approximation is positive definite, whereas the real NTK takes on negative values. In this section, we develop a better local approximation that overcomes these limitations. Our approach has the additional benefit that we may predict the correlation length-scale, along with other order parameters.

Our starting point is the local structure of the Cos NTK. The full details of our derivation are found in Appendix A.3, but the main strategy is to leverage the law of cosines to express the $C_{NTK}$ as:

$$C_{NTK}(x, x + u) = \frac{||\nabla_\theta f(x)||^2 + ||\nabla_\theta f(x + u)||^2 - ||\nabla_\theta f(x + u) - \nabla_\theta f(x)||^2}{2||\nabla_\theta f(x)|| \, ||\nabla_\theta f(x + u)||} \quad (14)$$

Performing a Taylor expansion in $u$ and retaining terms up to second order, we find the Cosine NTK locally takes the form of a Cauchy Distribution:

$$C_{NTK}(x, x + u) \approx \frac{2a_x^2 + u^\top D_x}{2a_x^2 + u^\top D_x + u^\top H_x u}, \quad (15)$$

where we have:

$$a_x = ||\nabla_\theta f(x)|| \quad (16)$$

$$D_x = \nabla_x ||\nabla_\theta f(x)||^2 \quad (17)$$

$$H_x = (\nabla_x \nabla_\theta f(x))(\nabla_x \nabla_\theta f(x))^\top \quad (18)$$

To obtain a correlation length-scale from this anisotropic model, we note that the level sets of equation 15 correspond to ellipses. For a given value $c$, the area of the level set can be shown to be (see Appendix A.4):

$$A_{ellipse}(x; c) = \frac{\pi}{\sqrt{\det H}} \left( \frac{2(1 - c)}{c} a_x^2 + \frac{(1 - c)^2}{4c^2} D^\top H^{-1} D \right) \quad (19)$$

To take into account the asymptotic value of $C_{NTK}$, we choose $c = 1/2 + c_\infty/2$. Then:

$$\xi(x) \approx \sqrt{\frac{A_{ellipse}(x; 1/2 + c_\infty/2)}{\pi}} \quad (20)$$

## 3.3 ORDER PARAMETERS FOR THE ONSET OF NTK ALIGNMENT

In the classification problems typically studied in the NTKA literature, the principle eigenvector $v_0$ is seen to learn class-separating boundaries (Kopitkov & Indelman, 2020; Baratin et al., 2021). Similarly, for our 2D image reconstruction task, we see the NTK learns information about the distribution of

edges in the image (Figure 4). To quantify this alignment, we use a Canny Edge Detector (Canny, 1986) to estimate connected image edges. We then quantify the utility of $v_0$ in predicting edges in terms of average recall, as measured by the area under the Receiver Operating Characteristic Curve (ROC AUC). We denote this measure $\text{AUC}(v_0, \nabla I)$, and it has the advantage of being insensitive to monotonic transformations of $v_0$.

Another hallmark of NTKA is early anisotropic growth of the spectrum of the NTK (Baratin et al., 2021), as the NTK becomes stretched along a relatively small number of directions that are correlated with the task. This is especially the case for the principal eigenvalue $\lambda_0$, which becomes orders of magnitude larger than the next leading eigenvalue. In subsequent sections, we will demonstrate, empirically, that this also holds during INR training.

The divergence of $\lambda_0$ enables a particularly simple approximation of the princpal eigenvector $v_0$. Namely, because the principal eigenvalue is so dominant, $K_{NTK}$ becomes effectively low-rank, and so power iterations converge quickly. Thus, choosing a vector of ones $v = 1$ as our initial vector, we expect $K1/1^\top 1$ to have strong cosine alignment with the principal eigenvalue. In the continuum limit, this is simply given by:

$$K1/N \rightarrow \mathbb{E}_u[K(x, x + u)] \tag{21}$$

$$= \mathbb{E}_\epsilon[\mathbb{E}_u[K(x, x + u)| \, ||u|| = \epsilon]] \tag{22}$$

$$= \int_0^{\epsilon_{max}} d\epsilon \, k(x, \epsilon)P(x, \epsilon) \tag{23}$$

Here, $P(x, \epsilon)$ denotes the density of points that are located a distance $\epsilon$ from the point $x$, and $\epsilon_{max}$ is an upper bound on the distance that we assume is much greater than $\xi_{corr}$. Close to this $x^2$, $P(x, \epsilon)$ grows like $2\pi\epsilon$. Thus, leveraging equations 9 and 16, we have:

$$v_0(x) \approx 2\pi a_x^2 \int_0^{\epsilon_{max}} d\epsilon \, \epsilon \left[ c_\infty(x) + (1 - c_\infty(x))e^{-\epsilon^2/2\xi^2(x)} \right] \tag{24}$$

$$= 2\pi a_x^2 \left[ c_\infty(x)\epsilon_{max}^2 + \xi^2(x)(1 - c_\infty(x))(1 - e^{-\epsilon_{max}^2/2\xi^2(x)}) \right] \tag{25}$$

$$\approx a_x^2 \left[ c_\infty(x)\text{Vol}(\mathcal{D}) + 2\pi\xi^2(x)(1 - c_\infty(x)) \right] \tag{26}$$

As we approach the phase transition, the asymptotic values tend towards 0, and the second term dominates. Considering the approximation in equation 20 for the correlation length-scale $\xi$, we note that $v_0(x)$ grows as $\mathcal{O}(||\nabla_\theta f(x)||^4)$. This implies particular sensitivity to pixels in regions with substantial high-frequency information, such as edges and corners. As natural images tend to be piecewise smooth, pixels on boundaries have the strongest spatial gradients within their neighbourhood, and are therefore the greatest source of information, being poorly compressible due to the lack of smoothness/redundancy, and accordingly disagreement in paramater gradients. Given the inability of models to accurately describe sharp discontinuities these edge pixels can be considered as influential datapoints, which accounts for their prominence within the principal eigenvector. We consider parallels between these observations of the NTK principal eigenvector and traditional approaches from the image processing literature concerning corners and edges in Appendix E. The fidelity of our approximation is evaluated in Appendix F.

### 3.4 ORDER PARAMETERS FOR THE LOSS RATE COLLAPSE

In (Fort et al., 2020b), (Sankararaman et al., 2020), (Feng & Tu, 2021b), and related works, the authors examine the role of gradient alignment statistics in determining the speed of learning under stochastic gradient descent. They note that the emergence of negative alignments between batches correlates with a reduction in learning speed. Intuitively, when gradient alignment becomes negative, the sum of the gradients approaches zero, resulting in a diminished learning signal. The minimum alignment between the gradients is simply given by the minimum value of the Cos NTK, which we may obtain explicitly from equation 15 as follows (full derivation in Appendix A.5):

$$\min_u C_{NTK}(x, x + u) = \frac{D_x^\top H_x^{-1} D_x}{D_x^\top H_x^{-1} D_x - 8a_x^2} \tag{27}$$

---

[2]The true form of $P(x, \epsilon)$ is complicated and varies from point to point, due to edge effects. However, these effects are suppressed as $P(x, \epsilon)$ only appears when multiplied the Gaussian $k_x$.

$\min C_{NTK}$ is then simply the minimum of 27 across the whole dataset.

## 3.5 MAG-MA: ORDER PARAMETERS FROM TRANSLATIONAL SYMMETRY BREAKING

While previous sections have focused on a bottom-up construction and analysis of order parameters, this section adopts a top-down approach rooted in symmetry principles. In Sections 3.1-3.4, we expressed several order parameters in terms of the parameters $a, D, H$, which characterize the local structure of the $C_{NTK}$. Tellingly, each of these parameters is now a function of the spatial variation of the parameter gradients. This suggests it is a translation symmetry which is broken at the phase transition. Indeed, from Figure 3, we observe that the $C_{NTK}$ is an approximately stationary, isotropic kernel - a desirable property for INRs (Tancik et al., 2020). Phrased another way, the Kernel exhibits no bias for location or direction. Over the course of training, we may monitor the emergence of such a bias with the following metric :

$$||\mathbb{E}_x[\nabla_x \log ||\nabla_\theta f||^2]||^2 = ||\mathbb{E}_x[D_x/a_x^2]||^2 \tag{28}$$

We refer to this statistic as **MAG-Ma**: the **M**agnitude of the **A**verage **G**radient of the Log Gradient-Field **Ma**gnitudes. Intuitively, this order parameter captures the statistical preference for a spatial direction in the dataset. The evolution of this quantity is plotted in Figure 3, and its alignment with the other order parameters is shown in Figure 4. We see that throughout the Fast Phase of training (before the peak in the loss rate $\dot{L}_e val$), the local structure of the $C_{NTK}$ is statistically translation invariant, and MAG-Ma is close to zero. However, just after the critical point, it grows rapidly - coinciding with the structure learning described in Section 3.3.

## 4 EXPERIMENTAL RESULTS

### 4.1 EXAMINING THE DISTRIBUTION OF CRITICAL POINTS

In this section, we show that the critical points defined in Section 3 cluster around a common time. We train a range of SIREN models on a super-resolution task, using fifteen images of varying complexity. We vary width, depth, and the bandwidth $\omega_0$ which multiplies the pre-activations in SIRENs (Appendix B). We illustrate the results of this sweep in Figure 4. We also consider three more order parameters from the literature, which may be defined in terms of the NTK:

- We track the principal eigenvalue $\lambda_0$ of the NTK.
- In (Shwartz-Ziv & Tishby, 2017) and others, the authors consider the impact of the norm and standard deviations of model parameters on the loss rate. During the fast learning phase, the means of model gradients are large and the variances are small, with the converse true in the slower phase. In terms of the NTK and the residual $r$, the variance is (Appendix A.6):

$$\sigma_\theta^2 = \frac{1}{N}\text{Tr}(\text{diag}(r)^2 K_{NTK}) - \frac{1}{N^2}r^\top K_{NTK} r \tag{29}$$

- **Centred Kernel Alignment (CKA)**: Empirically, as a DNN learns features that support the prediction of a target, its NTK begins to resemble the task kernel $K_Y$. For INR regression, we opt to use $K_Y(x, x + u) = \exp(-||I(x) - I(x + u)||^2/2\kappa^2)$. Here, $\kappa$ is a bandwidth parameter (full details Appendix C.1). The similarity between kernels is measured using the normalized Hilbert-Schmidt Information Criterion (HSIC), as in (Baratin et al., 2021; Shan & Bordelon, 2022; Canatar & Pehlevan). We also measure alignment between the NTK and an RBF $K_X(x, x + u) = \exp(-||u||^2/2\kappa^2)$.

The left side of Figure 4 illustrates our procedure for identifying critical points. We use a simple peak detector to identify the region of interest for the loss rate $\dot{L}_{eval}$ and the gradient variance $\sigma_\theta$, using the FWHM to define a confidence region. For the $\min C_{NTK}$, we look for zero-crossings, with a confidence region constructed from the cumulative variance. For every other order parameter, we fit a sigmoid, where the inflection point marks the critical point, and the slope defines the confidence region (full details in Appendix C.1). The right side of Figure 4 demonstrates how frequently these confidence regions overlap across the different architectures and images studied[3]. Remarkably, the

---

[3] In computing the coincidence matrix on the right side of Figure 4, we only included experimental runs in which critical points were detected for both pairs of order parameters. In Appendix C, we comment on how image properties impact the detection rate.

phase transitions described by the order parameters - despite being derived to measure different phenomenon in the literature - consistently occur at the same time during training.

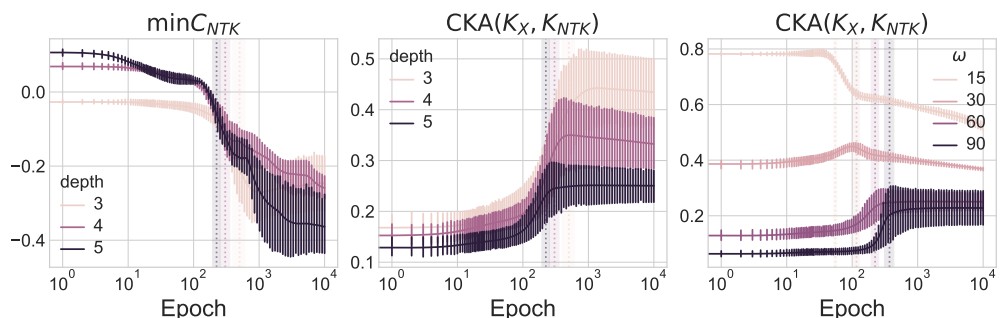

Figure 4: **Alignment of Order Parameters**. Left: Order parameter evolution and critical points during training of a SIREN model on the astro image. The red vertical lines denote the location of the critical points, and the green vertical lines denote confidence regions. Right: Heatmap showing the frequency of intersections between the confidence regions. Additional figures in Appendix G.1.

Figure 5: **Effect of Hyperparameters on Critical Behaviour**: Average MSEs, in order of ascending depth: $7.742e^{-3} \pm 1.580e^{-4}$, $6.819e^{-3} \pm 2.696e^{-5}$, $6.571e^{-3} \pm 2.705e^{-5}$. Average MSEs, in order of ascending $\omega_0$: $8.380e^{-3} \pm 9.191e^{-5}$, $7.234e^{-3} \pm 5.591e^{-5}$, $6.571e^{-3} \pm 2.705e^{-5}$, $7.853e^{-3} \pm 5.629e^{-4}$. Dashed vertical lines denote the location of the peak of the loss rate $\dot{L}_{eval}$, marking the phase transition. Additional figures in Appendix G.2.

## 4.2 DYNAMICAL CONSEQUENCES OF HYPERPARAMETERS

In this section, we perform an ablation study to understand the impact of different hyperparameters on the order parameter trajectories. The baseline model is a 5-layer 128-unit wide SIREN with $\omega_0 = 60$, which on average was the best performing model on the cameraman dataset. We visualize the evolution of $\min C_{NTK}$ in Figure 5, in addition to kernel alignment with a static RBF. Error bars are obtained by averaging the runs over five random seeds.

When depth (and therefore model capacity) is decreased, we observe a corresponding increase in the validation error. In shallower models, the initial $\min C_{NTK}$ is lower, delaying learning, and thus,

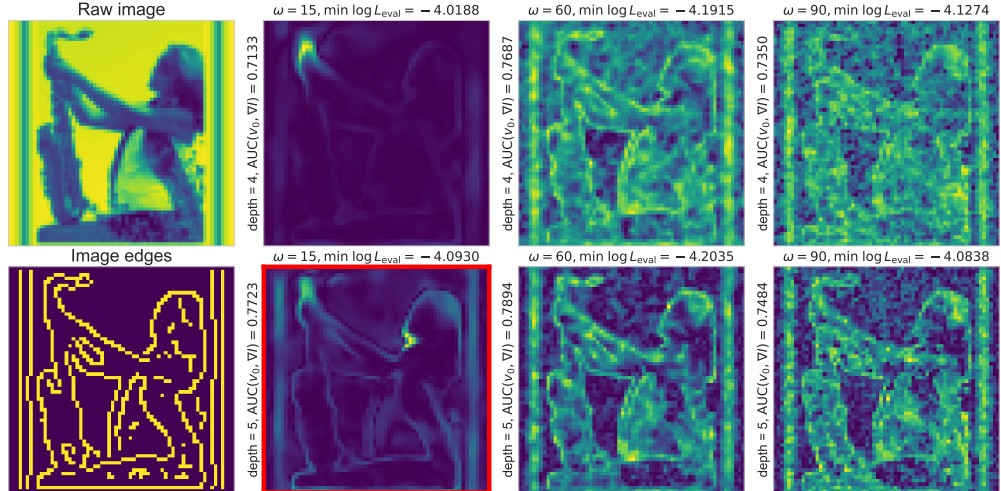

Figure 6: **Effect of Hyperparameters on Edge Alignment**: Visualizing the principal eigenvector of the NTK for models trained with different hyperparameters. Best performing model (by validation loss) highlighted in red. More images in Appendix C.2

|          | $\omega_0 = 15$ | $\omega_0 = 30$ | $\omega_0 = 60$ | $\omega_0 = 90$ |
|----------|-----------------|-----------------|-----------------|-----------------|
| depth = 3 | $0.557\pm 0.039$ | $0.549\pm 0.057$ | $0.557\pm 0.051$ | $0.550\pm 0.041$ |
| depth = 4 | $0.663\pm 0.077$ | $0.698\pm 0.085$ | $0.675\pm 0.075$ | $0.634\pm 0.069$ |
| depth = 5 | $0.708\pm 0.078$ | $\mathbf{0.732 \pm 0.081}$ | $0.709\pm 0.079$ | $0.669\pm 0.071$ |

Table 1: **Variation in** $\mathrm{AUC}(v_0, \nabla I)$ **with hyperparameters**: Maximum values of the edge alignments reached during training, averaged across all runs with fixed depth and $\omega_0$. Highest average edge alignment indicated in bold. Values computed over the same sweep defined in Section 4.1.

the peak in the loss rate $\dot{L}_{eval}$. While the location of the phase transition changes, the trajectory of the order parameters shapes remain consistent, and exhibit less variance with increased depth. By contrast, there is dramatic change in the shape of the trajectories as we vary $\omega_0$. When $\omega_0$ is high, $\xi_{corr}$ starts very low, favouring interactions with immediate neighbours, leading to low overlap with the RBF. During training, the range broadens rapidly, causing $\mathrm{CKA}(K_X, K_{NTK})$ to grow sigmoidally at the critical point. Conversely, with low $\omega_0$, the range starts large but shrinks during training.

## 4.3 Influence of Hyperparameters on Edge Alignment

Generally, we find that the principle eigenvector $v_0$ exhibits the sharpest alignment with edges in deeper models (Table 1) that are trained on more complex images (Figure 10 in Supp. Materials). To gain intuition, we perform an ablation study to understand how $\mathrm{AUC}(v_0, \nabla I)$ varies with depth and $\omega_0$ in Figure 6. We use the same baseline model described in Section 4.3 on the sax dataset. Results on additional datasets can be found in Appendix C.2. We make two observations:

1. Increasing $\omega_0$ initially increases $\mathrm{AUC}(v_0, \nabla I$ as more complex structures appear, but this falls for higher $\omega_0$. One interpretation is that, similar to an RBF, the preference for local interactions induced by the higher $\omega_0$ causes the model to overfit. From Section 3.3, we know that $v_0(x) \sim \mathcal{O}(||\nabla_\theta f(x)||^4)$, which grows for the points where the model has overfit. This obscures the edges, leading to lower $\mathrm{AUC}(v_0, \nabla I)$. The overfitting perspective also clarifies why NTK alignment tends to co-occur with the fast-slow learning transition, which has been identified with the onset of memorization (Shwartz-Ziv & Tishby, 2017).

2. As we increase depth, the principal eigenvector becomes sparser, concentrating on the edges, and ignoring background pixels, even with higher $\omega_0$. This can also be explained by the gradient mag-

nitudes, which for deeper models decompose as $||\nabla_\theta f||^2 = \sum_{l=1}^{\text{depth}} ||\nabla_{\theta^{(l)}} f||^2$. Thus, preference is given to points which are consistently confusing across layers, mitigating the effects of noise.

## 5 RELATED WORK

**Neural Tangent Kernels for Implicit Neural Representations**: Previous research has investigated the inductive biases of INRs using the Neural Tangent Kernel (NTK), focusing on aspects such as spectral properties (Li et al.) and dependencies on uniformly sampled data (Tancik et al., 2020). Furthermore, studies by Yüce et al. (2022) and Saragadam et al. (2023) have analyzed the eigenfunctions of the empirical NTK to elucidate the approximation capabilities of INRs. These investigations, however, primarily examine static properties of the NTK at initialization, which do not account for feature learning dynamics. In contrast, our work concentrates on the evolution of the NTK, aiming to deepen our understanding of how INRs learn to model images.

**Neural Tangent Kernel Alignment** In practical settings, recent studies have shown that during training, the NTK dynamically aligns with a limited number of task-relevant directions (Fort et al., 2020a; Geiger et al.; Kopitkov & Indelman, 2020; Paccolat et al.; Baratin et al., 2021; Atanasov et al., 2021; Shan & Bordelon, 2022; Canatar & Pehlevan). Concurrently, at the eigenfunction level, the modes increasingly reflect salient features of the dataset, such as class-separating boundaries (Kopitkov & Indelman, 2020; Baratin et al., 2021). The widespread occurrence and influence of kernel alignment suggest its critical role in DNN feature learning, contributing to the superior performance of DNNs over models based on infinite-width NTKs (Shan & Bordelon, 2022). That said, these theoretical discussions often focus on shallow networks (Paccolat et al.; Atanasov et al., 2021), toy models Shan & Bordelon (2022); Baratin et al. (2021), and deep linear networks (Atanasov et al., 2021). In contrast, the INRs we study are deep, nonlinear models.

## 6 CONCLUSION

We have conducted preliminary investigations into the dynamics of feature learning within SIRENs for image data. By analytically deriving approximations for the local structure of SIREN NTKs - using Gaussian and Cauchy distributions - we were able to obtain approximate expressions for the correlation lengthscale, the minimum value of the $C_{NTK}$, and the principal eigenvector. We related these expressions to order parameters for three phase transitions identified in different dynamical perspectives on learning: the appearance of diffusion wave-crests in residual evolution (first identified in this paper); the collapse of the loss rate; the onset of NTK alignment. We argued, based on these derivations and empirical demonstrations that critical points cluster in time, that these distinct phase transitions share a common, underlying mechanism.

The following picture emerges from our analysis: as long range correlations between gradients decay, residuals only interact with their immediate neighbours (onset of diffusion), leading to increased gradient variance (loss rate collapse) and translational symmetry breaking. In parallel, the growth of the principal eigenvalue or the NTK leads the principal eigenvector to memorize the distribution of influential points, as measured by accumulating gradients. In images, one influential class of points are edges, leading to their prominence in the principal eigenvector (NTK alignment).

Overall, many promising lines of research remain open. In this study, we focused on SIREN models trained on a 2D super-resolution task using full-batch gradient descent. However, SIRENs are used in a variety of inverse problems, and it remains to be seen whether our observations extend to these settings. Future work may also explore the impact of different optimizers, such as ADAM (Kingma & Ba, 2014), which adaptively adjusts learning rates and may influence the stability and divergence of the principal eigenvalue - a key factor in our study of NTK alignment. Finally, while the NTK is computationally expensive to evaluate, limiting its practical use for monitoring models, MAG-MA is efficient to compute and may offer promising applications worth further investigation.

This work has demonstrated how the NTK provides a rich theoretical tool for deriving and relating order parameters to understand training dynamics. Our approach provides new methodology to rigorously study the influence of inductive biases, such as model architectures and hyper-parameter values, on the underlying learning process and may have practical utility in diagnosing the cause of poor learning outcomes.

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
