# OpenReview forum: "Common Causes for Sudden Shifts: Linking Phase Transitions in Sinusoidal Networks"
_ICLR.cc/2025/Conference — Submitted to ICLR 2025_

### Official Review · Reviewer_s3fX · 2024-10-31

**Soundness:** 3
**Presentation:** 2
**Contribution:** 4
**Rating:** 6
**Confidence:** 4

**Summary:**

This paper investigates the role of phase transitions during the training of Implicit Neural Representations. More specifically, it is revealed that phase transitions occur simultaneously. For this reason, three lenses are considered: spatial distribution of residuals, principal eigenvectors of the NTK, and training curve analysis. The first achievement is a unification of the different associated order parameters, which actually provides a theoretical reasoning for the occurrence of the phase transitions. The second major achievement is the introduction of a novel parameter termed MAG-Ma explicitly tracking violations in the translational symmetry of the NTK. In several numerical experiments, the distribution of critical points and the dynamics of hyperparameters are examined.

**Strengths:**

Based on my understanding, the paper offers several novel and insightful contributions, such as the identification of phase transitions in clearly defined periods and the introduction of the MAG-Ma parameter. These findings are partially rigorously justified, and the main assertions are also partly supported by numerical verification. If these findings are correct, this paper would represent a significant advancement in the theory of sinusoidal networks.
The numerical experiments impressively show the alignment of the critical points of the order parameters and the effect of depth on the critical behavior.

**Weaknesses:**

This paper suffers from several major drawbacks.
First, many central statements only hold true approximately (excluding equations 10 and 11, which are numerically verified). For instance, I cannot validate the quality of the approximations in (13), (14), (28), or (29), which are crucial for major findings in the manuscript. Here, either additional numerical experiments or theoretical justifications would be highly appreciated.
Second, the presentation of the paper should be improved. Indeed, some parts of the paper can only be understood after reading the supplementary material. However, the authors do not mention the additional proofs in the supplementary material in the main manuscript, which could be tackled by forward references to the supplementary material.
I am not sure if the numerical results in Figures 4-6 only hold true in general or only for specific situations. Additional numerical experiments would be required since these findings are crucial for the correctness of the entire approach.
Finally, I am not sure about the simultaneity of the order parameters since according to Figure 4, the confidence regions of MAG-Ma and many other order parameters hardly overlap. In this sense, MAG-Ma can be interpreted as a time-shifted order parameter.


Some minor remarks:
- typo "equation equation 5" on page 3
- "Which resembles" instead of "which resembles" on page 4
- "Where we have" instead of "where we have" on page 5
- please cite the "related works" on page 6

**Questions:**

1) Can you please either provide rigorous mathematical justifications or additional numerical experiments for the approximations mentioned above? I would like to be sure about the correctness of all these statements.
2) From my perspective, additional numerical experiments for Figures 4-6 are required to disprove that the results are only valid for this specific configuration. Could you please perform these experiments on other datasets or in modified settings?
3) Is MAG-Ma a time-shifted order parameter as suggested in Figure 4 or is this only true for this specific experiment? Can you please provide additional numerical experiments for this temporal evolution? Is this shift consistent across different experimental settings? Do you have any theoretical justification?
4) To improve the presentation, I would like to have more forward references to the relevant sections in the supplementary material whenever possible and reasonable.

---

> ### Author Response · Authors · 2024-11-20
>
> We are very grateful for the detailed review. We have the following comments:
>
> > For instance, I cannot validate the quality of the approximations...either additional numerical experiments or theoretical justifications would be highly appreciated.
>
> In our original submission, we numerically evaluated some of our approximations in Section D of the supplementary materials (now called Section E in light of other changes).  We should have pointed to this section more explicitly (we now do this at the end of Section 3.3).  We have also added more numerical experiments to this section to help validate the quality of our approximations (ex gauging the cosine alignment between the true principal eigenvector and our approximation from Section 3.3).
>
> > Finally, I am not sure about the simultaneity of the order parameters since according to Figure 4, the confidence regions of MAG-Ma and many other order parameters hardly overlap. In this sense, MAG-Ma can be interpreted as a time-shifted order parameter.
>
> Thank you for the feedback.  We believe this was an error in how we were performing change detection for MAG-Ma.  In the symmetric phase, the MAG-MA statistic is close to zero, and at the phase transition, it grows rapidly.  To identify the critical point, as we did for CKA, AUC, and other parameters, we fit a sigmoid function to the trajectory of the order parameter.  The inflection point $t^*$ of the sigmoid then corresponds to the critical point, with the slope $w$ providing a confidence interval $[t^*-2w, t^*+2w]$ (by analogy with the Gaussian erf, this corresponds to the region with $2\sigma$ certainty).
>
> The issue with this, of course, is that, for MAG-Ma, that inflection points happens well after the statistic has grown past zero!  To this end, it makes more sense to use the interval $[t^*-3w, t^*-w]$ for MAG-Ma.  With this change, the confidence interval for MAG-Ma cleanly overlaps with the other order parameters, as seen in Figure 4.
>
> > From my perspective, additional numerical experiments for Figures 4-6 are required to disprove that the results are only valid for this specific configuration. Could you please perform these experiments on other datasets or in modified settings?
>
> We have expanded the supplementary materials with additional figures to show that the findings indeed hold across a number of settings.  Though the specific shapes and timings vary, the observations described in Section 4.1 and 4.2, pertaining to how the shapes and timing vary with hyperparams, hold across datasets.

---

> > ### Comment · Reviewer_s3fX · 2024-11-20
> >
> > Thanks for the update! I am not enthusiastic about the paper, but given your precise answers I changed my rating.

---

### Official Review · Reviewer_r32t · 2024-11-02

**Soundness:** 3
**Presentation:** 3
**Contribution:** 2
**Rating:** 5
**Confidence:** 3

**Summary:**

Submission 957 investigates the dynamics of SIREN training. SIREN is an MLP (implicit neural representation) with sinusoidal activations often trained to solve tasks on low-dimensional domains such as fitting images, solving PDEs, etc.

In practice, SIRENs often struggle early in training and then suddenly fit the image, corresponding to a sharp drop in loss (a “phase transition”). Submission 957 presents an analysis using neural tangent kernel (NTK) machinery to correlate empirical phenomena at the transition to changes in the NTK.

**Strengths:**

- The empirical finding that the translational symmetry of (the local approximation of the) NTK is broken at the phase transition is both novel and interesting.
- The paper’s presentation is broadly accessible to practitioners (such as myself) familiar with the NTK and SIREN literature but not with more recent theoretical advancements. This is a non-trivial strength for papers in the learning theory track.

**Weaknesses:**

I’m happy to reconsider my score during the discussion so I look forward to the rebuttal. As of now, I see the following weaknesses:

## Unclear theoretical contributions:

The paper’s main strategy is to observe empirical phenomena in SIREN training (e.g., ripples in reconstruction, the image suddenly being fit, sharp drops in loss) and try to correlate them with changes in the NTK (or a local approximation thereof). However, these analyses all appear to be speculative and do not explain _why_ these phenomena occur.

For example, Section 3.1 presents an analysis of why ripples appear during SIREN training. However, the analysis is based on fitting isotropic Gaussians to key parameters which the paper itself states (in Sec 3.2) is not appropriate and disconnected from actual empirical training. Further, the paper’s arguments for symmetry breaking appear to contradict this assumption as well. It is thus unclear what to take away from the claim that this analysis explains the ripples.

Looking holistically at Section 3, the paper shows that when training starts to work (at the “phase transition”), the NTK changes. However, please correct me if I’m wrong, but this is tautologically true – the NTK always changes when training starts to work. For example, the introduced metric $AUC(v_0, \nabla I)$ just plainly tracks when image edges start to align with the empirical NTK eigenvector so the argument comes down to “training works when the image edges start appearing” which would also correlate one-to-one with loss, PSNR, SSIM, etc. Please correct me if I’m missing something.

The actual finding relevant to SIRENs (translational symmetry breaking) in Section 3.5 is entirely empirical. Of course, this empirical insight is valuable as well, but I don’t know what to make of it as it does not explain _why_ it breaks. Further, it is unclear what aspect of the theoretical analyses is specific to sinusoidal networks/SIREN and not any other INR parameterization. As far as I could tell, it essentially comes down to fitting a Cauchy distribution to the empirical CosNTK of a SIREN. In future revisions, please disambiguate which aspects are specific to SIREN.

Importantly, all of the empirical phenomena upon which theoretical arguments are made are only drawn from 2D image fitting and the other use cases of INRs are not addressed. It is hard to see how the presented rationale would generalize to entirely different contexts, for example, inverse problems such as MRI or CT reconstruction where INRs are also widely used.

Lastly and importantly, there is no theoretical analysis of the frequency aspect of INR training (as in the NTK-based [Fourier features paper](https://arxiv.org/pdf/2006.10739) ). SIREN training depends entirely on the $w_0$ parameter and initialization, but none of the analysis in Section 3 accounts for it and thus it provides an incomplete picture of SIREN dynamics. In fairness, the experiments do sweep $w_0$ and do find different behaviors, but that precisely shows that a theoretical analysis is needed.

## Non-rigorous experiments:

As rigorous theoretical explanations are not presented for these phenomena, we turn to the experiments. However, the experiments are far too preliminary in the current submission to have clear takeaways:
- Again, there is only a single problem studied (2D image fitting), whereas previous analyses of INRs study a broad range of low-dimensional domain problems. For example, SIREN studies waveform inversion, Poisson image reconstruction, signed distance function fitting, etc. The closely related NTK-based Fourier features paper studies a variety of MRI/CT reconstruction and regression problems as well.
- Even within the context of 2D image fitting, the experiments only study five images in total (e.g. cameraman). It is hard to gain generalizable insights from five toy images on a single task.


## Presentation issues:
- The entire preamble of Section 3 (+Fig 1) should moved to the Introduction as that is where the paper is actually outlined and the main contributions are clearly defined. I would merge it with the current contributions statement. Any additional space could then be allocated to further experiments.
- Section 3 begins with “Having outlined the connection between learning in INRs and field theory, ..”. What precedes it is just definitions, most of which are not specific to INRs, and using “field theory” to refer to spatial gradients/residuals/etc. is just confusing. Please refrain from borrowing terms from physics in this context.
- Minor presentation issues:
    - The writing in the title and abstract keeps implying multiple phase transitions, whereas there is only one. Please make this clearer.
    - typo: There’s no line numbers so control+f for “equation equation 5”
    - Page 3, section 3: Please explicitly refer to the subplots of Figure 1 in the text.

**Questions:**

- Do the theoretical arguments made for 2D image regression in this paper generalize to the wider set of tasks where INRs are used?
- Could the authors please clearly disambiguate the novel theoretical contributions of this work?
- What is the contribution of Section 3.1 if the rest of the paper contradicts it?
- Is there an argument for _why_ translational symmetry breaking or any of the other empirical phenomena are occurring?
- Given that SIREN training depends entirely on $w_0$, why was it excluded from analysis?
- Why are the experiments focused on regressing just five images? Also, are the experiments based on performance on test pixels or is the entire image used? If so, is there a train test gap in the theory and experiments?

Please see the weaknesses above for more.

---

> ### Author Response · Authors · 2024-11-20
>
> We are very grateful for the detailed review. We have the following comments:
>
> > However, these analyses all appear to be speculative and do not explain _why_ these phenomena occur.
>
> Thank you for the feedback. We agree that a full theoretical explanation of _why_ these phenomena occur is a complex undertaking and remains an open research question.  However, as with previous [ICLR submissions](https://arxiv.org/pdf/2103.00065), we believe the novelty of our analysis and observations is sufficient to inspire further research, even though we cannot, at present, provide a detailed accounting of their origins.
>
> > The analysis is based on fitting isotropic Gaussians to key parameters which the paper itself states (in Sec 3.2) is not appropriate and disconnected from actual empirical training. Further, the paper’s arguments for symmetry breaking appear to contradict this assumption as well.
>
> We believe we can state the main contribution of 3.1 more clearly. The key point of this section is to demonstrate how the experimentally-observed locality of the SIREN NTK (as visualized in Figure 2), when analysed in the continuum, leads naturally to the wavelike evolution of the residuals.  The diffusion equation merely offers us some theoretical justification for our choice of orders parameters (the asymptotic value $C_{\infty}$ of the $C_{NTK}$, along with the correlation length scale $\xi_{corr}$).
>
> We initially chose the uniform Gaussian approximation to simplify our presentation, as a means to build intuition about how strongly-local kernels result in diffusion equations.  We have now updated our derivation in Section 3.1 to include a spatially-varying correlation lengthscale $\xi$ (the form of the diffusion equation is mostly unchanged).  Note that it is possible to carry out the derivation with a anisotropic Gaussian as well.  The main difference is that the simplification in equations 50-51 is not possible, but the resulting equation (55) would still be a diffusion equation, just an anisotropic one (the second order term would look like $\text{tr}(\Sigma(x)\nabla_x^2 r(x))$).
>
> We believe this addition is unnecessary, however, as in Section 3.2, we refine the local approximation to encompass other spatial variations in the NTK.  Likewise, our discussion on symmetry breaking does not refute the assumption of locality: we simply observe that, at the beginning of training, there is very little spatial variation in the local structure of the NTK, whereas at the phase transition, we observe significant increase in spatial variation.
>
> > However, please correct me if I’m wrong, but this is tautologically true – the NTK always changes when training starts to work...the argument comes down to “training works when the image edges start appearing”...
>
> To clarify, it would be more correct to say that, as edges start appearing in the principal eigenvector $v_0(x)$, training begins to slow down.  Per the argument in Section 3.3, $v_0(x)$ grows as $\mathcal{O}(||\nabla_{\theta}f(x)||^4)$, which in turn grows for data points that are noisy/influential in training.  One candidate explanation is accumulating errors in modelling the edges leads to increased gradient confusion, which in turn slows down learning.  However, at present, the exact causal mechanism falls outside the scope of the paper (though some preliminary analysis is found in Section C).  What our paper does report is the surprising fact that these different dynamical characteristics exhibit sudden, simultaneous changes at a critical point during training.  As far as we can tell, this observation is not obvious, nor has it been reported before.  We would be very grateful if you could point us towards a reference.

---

> > ### Author Response · Authors · 2024-11-20
> >
> > > In future revisions, please disambiguate which aspects are specific to SIREN.
> >
> > Thank you for the feedback.  We have added a new section to the supplementary materials (section D) which compares SIREN models to ReLU-MLPs with a positional encoder, as used in [44].  Our main empirical takeaway is as follows: while ReLU-PE models also exhibit Neural Tangent Kernel alignment, it is a much slower, non-local process, that does not coincide with loss-rate collapse or translational symmetry breaking.  In contrast, in SIRENs, the NTK is dominated by its local structure, and NTK alignment takes place alongside loss rate collapse, translational symmetry breaking, and the onset of diffusion.
> >
> > > It is hard to see how the presented rationale would generalize to entirely different contexts, for example, inverse problems
> >
> > Our validation set is a super-resolution task.  We described this in Section B.1 of the supplementary materials, but we should have made this more explicit in the main body of the paper.
> >
> > > Lastly and importantly, there is no theoretical analysis of the frequency aspect of INR training
> >
> > We appreciate the reviewer's insight regarding the importance of a theoretical analysis of the frequency aspects of INR training. While a comprehensive spectral analysis is indeed valuable, it lies outside the scope of this work, which focuses on the novel investigation of spatio-temporal residual dynamics. Existing literature already extensively covers the spectral/frequency analysis of SIRENs and other INRs (e.g., the [Fourier features paper](https://arxiv.org/pdf/2006.10739)). Our work complements these studies by exploring the spatial structures in the residuals, and the eigenvectors of the NTK.  We believe this novel contribution provides significant insights into SIREN training dynamics and justifies the focus of our current investigation.
> >
> > > Importantly, all of the empirical phenomena upon which theoretical arguments are made are only drawn from 2D image fitting and the other use cases of INRs are not addressed.
> >
> > We appreciate the reviewer's feedback. It's important to note that this work represents a significant advancement in the complexity of models studied within the Neural Tangent Kernel Alignment (NTKA) literature. Previous NTKA analyses frequently focus on classification tasks with simpler architectures, such as linear networks (references [23,24,35]). Our investigation of INRs, which, as deep MLPs, are substantially more complex, represent a significant step forward in the study of this phenomenon. While our empirical analysis focuses on 2D image fitting to manage this increased complexity (specifically analysing the evolution of the residuals), we would be very excited to adapt our analysis to other tasks in the future.
> >
> > > The experiments only study five images in total
> >
> > We have included ten additional images (for a total of fifteen datasets) in our revision, with many additional figures now present in the supplementary materials to demonstrate the generality of our results.
> >
> > > Presentation issues
> >
> > We appreciate all of the the reviewer's feedback regarding the writing and presentation.  We've removed many of the references to field theory in the discussion of preliminary materials, though we retain it for the discussion of correlation functions.

---

> > > ### Comment · Reviewer_r32t · 2024-11-25
> > >
> > > I thank the authors for their thoughtful and detailed responses. Could the authors highlight all changes in the submission and supplement in a different font color, please?
> > >
> > > I fully agree that scientific work (i.e., finding and modeling training dynamics) is of major importance. What I'm stuck on is whether there are robust generalizable findings (whether theoretical or empirical) with respect to SIREN MLPs within the current version submission.
> > >
> > > The paper currently studies training on 15 64x64 images and tracking reconstruction error on 256x256 versions of the same (the original submission had 5). Within this context, I have the following questions/comments:
> > > - **Inconsistent results in a small test set:** As in Appendix C, we see that many of the purported phase transitions do not actually occur a substantial number of times, and whether or not they do occur depends on the image itself. I do not see how we can make confident statements about training dynamics if (a) the considered N is this low and (b) if the trends are inconsistent even within N=15. Is the main takeaway that SIRENs struggle to fit images with sharp spatial gradients?
> > > - **Tasks beyond image regression:** The point regarding other tasks beyond image regression was not adequately addressed in the rebuttal. SIREN has unique benefits over ReLU+PosEnc in other problems where higher-order gradients are required (e.g. solving 2nd-order differential equations as in the original SIREN paper) and it is in scope IMO to see whether the proposed phenomena occur in these other tasks as well. The findings reported in the original SIREN paper (or similarly, the Fourier Features paper) are demonstrated to be consistent across multiple tasks and not just one.
> > > - **Potential test set leakage:** In the rebuttal, it is clarified that the considered task is training on 64x64 and testing on 256x256. Please correct me if I'm wrong, but does this not then create "test set" leakage as the same data points were observed in both cases (albeit at different resolutions)? As far as I am aware, generally, INRs in this subdomain are trained on one set of pixels and tested on an offset grid which is never sampled during training (see eg [Appendix E.1](https://arxiv.org/pdf/2006.10739) ) such that we have some confidence that our evaluation is not overfitting to the observed measurements. Could you please elaborate on this point?
> > >
> > > > What our paper does report is the surprising fact that these different dynamical characteristics exhibit sudden, simultaneous changes at a critical point during training. As far as we can tell, this observation is not obvious, nor has it been reported before. We would be very grateful if you could point us towards a reference.
> > >
> > > There's a mixup of our points. To clarify, I mean that if there is a critical point in training, it is unclear to me why we would expect these characteristics (which are correlated with image edges) to *not* all change at the same time coinciding with the appearance of edges in the reconstruction and why this is a non-obvious statement. It is entirely possible that I'm missing something deeper, so please let me know.
> > >
> > > > We have added a new section to the supplementary materials (section D) which compares SIREN models to ReLU-MLPs with a positional encoder, as used in [44]. Our main empirical takeaway is as follows: while ReLU-PE models also exhibit Neural Tangent Kernel alignment, it is a much slower, non-local process, that does not coincide with loss-rate collapse or translational symmetry breaking. In contrast, in SIRENs, the NTK is dominated by its local structure, and NTK alignment takes place alongside loss rate collapse, translational symmetry breaking, and the onset of diffusion.
> > >
> > > There's a mixup again, unfortunately. To clarify, I am not asking about ReLU-PE networks vs SIREN training dynamics in this context. The proposed analysis machinery in Section 3 does not seem depend on whether it is a SIREN model being studied, outside of the empirical distribution fitting upon which some of the analysis is based. My question is about whether this approach is specific to SIREN in any way outside of the curve fitting.
> > >
> > > > While a comprehensive spectral analysis is indeed valuable, it lies outside the scope of this work, which focuses on the novel investigation of spatio-temporal residual dynamics. Existing literature already extensively covers the spectral/frequency analysis of SIRENs and other INRs (e.g., the Fourier features paper).
> > >
> > > I do not agree hat a spectral analysis is out of scope for this paper. In practice, $w_0$ is the most important hyperparameter to vary training dynamics and the properties of the reconstructions. I do not see how models of SIREN training dynamics can ignore the choice of $w_0$, especially as the experiments also empirically show how varying it changes the other identified order parameters.

---

> ### Author Response · Authors · 2024-11-27
>
> We would like to thank the reviewer for their thoughtful review.  As we understand it, the reviewer's primary concerns are:
> 1. Certain results in the paper seem to be obvious.
> 2. How applicable are our observations for the broader range of tasks that SIREN models are used for?
>
> We believe that both of these concerns are the result of a mixup regarding the goals of our paper.  For example, objection (1) seems to stem from the following:
>
> > ...if there is a critical point in training, it is unclear to me why we would expect these characteristics (which are correlated with image edges) to _not_ all change at the same time coinciding with the appearance of edges in the reconstruction and why this is a non-obvious statement.
>
> We would like to clarify that the critical point does not concern the appearance of edges in the **reconstruction** (ie the model outputs $f_{\theta}(x)$), but in the **principal eigenvector** $v_0(x)$ of the NTK (ie the Gram matrix for the gradients $\nabla_{\theta}f_{\theta}(x)$).  Whereas the former is directly trained to mimic a target signal, the latter is an emmergent property of the system, whose evolution during training is still an active area of investigation.
>
> Here's why this isn't obvious: In CNNs, where pixels correspond to features, feature-based interpretability tools highlight how the early layers of the network learn edge detectors.  In INRs, however, pixels correspond to individual datapoints.  Unfortunately, the highly nonlinear nature of gradient descent obscures the relationship between the final learned parameters of a DNN, and the dataset it was trained on.
>
> In the case of 2D image regression, we have some intuition about what datapoints are important, based on the wealth of Computer Vision literature.  The question is, can we demonstrate that these intuitions are valid for SIREN models?
>
> The NTK offers a simplified picture of dynamics under gradient descent, recasting evolution in terms of the residuals, instead of the model's parameters.  There are, however, many gaps here, chief amongst them being the gap between static NTK approximations (which empirically describe SIRENs very poorly, [see Figure 4 here](https://arxiv.org/pdf/2410.21645v1)) and the true dynamics of the network.  In practice, the dynamics of the empirical NTK, and its principal eigenvectors, is a topic of active research, though many theoretical analyses focus on linear architecture, on a single task (classification) (Atanasov et al., (2021); Shan & Bordelon (2022); Baratin et al. (2021)).
>
> Our work is a significant step forward in that regards, because SIRENs (the focus of this paper) are fully-nonlinear DNNs.  In particular, we analyze:
> - **What** does the principal eigenvector learn?  Phrased another way, what is the NTK aligning with?  The empirical observation that the SIREN NTK is dominated by its local structure allows us to relate (in Section 3.3) the principal eigenvector to the distribution of edges in an image.
> - **When** does NTK alignment take place? Our theoretical and empirical studies demonstrate that NTK alignment, in SIREN models, happens suddenly, and around the same time that other phase transitions occur.
>
> We have reorganized our introduction in order to more clearly emphasize our focus.

---

> > ### Author Response · Authors · 2024-11-27
> >
> > With the above goals in mind, we believe that the most fair comparison of our work is not made against the literature on INRs, but on NTK alignment. This is why we think the reviewer's second objection is out of scope for this paper.  Indeed, as the reviewer notes:
> >
> > > The findings reported in the original SIREN paper (or similarly, the Fourier Features paper) are demonstrated to be consistent across multiple tasks and not just one.
> >
> > It is certainly an interesting question, worthy of subsequent works, to examine if any of the previously stated analyses carry forward to tasks beyond curve fitting in SIRENs.  However, this would require significant expansion of our analytical framework to be able to address this question.  For example, using a loss function for reconstruction of the Laplacian, instead of the original image, would necessitate studying terms like $\nabla_{\theta}(\Delta_x f_{\theta}(x))$ (these terms are the generalization of the residuals in regression).
> >
> > Likewise, accomodating other tasks experimentally would require significant expansion of our engineering framework.  For example, in Appendix A.1 of the Fourier Features paper, the authors acknowledge that certain NTK experiments are not tractable beyond the small 1D regression problems they study, and highlight other memory constraints.  Similarly, because of memory constraints, it is not feasible to explicitly construct the NTK for the higher dimensional tasks considered in these papers - which we would need to evaluate our order parameters (the main topic of this paper).
> >
> > Finally, the SIREN and Fourier Features paper are principally concerned with designing novel architectural components for use in general INR tasks.  Thus, they choose multiple tasks in order to demonstrate the utility of these components, even though the use cases deviate significantly from their analyses.  By contrast, our contributions are primarily theoretical, meant to advance the literature on NTK alignment, and the case considered in our paper already constitutes a novel extension.  We note that other works studying the NTKs of SIRENs (such as [link](https://arxiv.org/abs/2112.01917)) have also only considered the case of 2D reconstruction.
> >
> > We raise the same argument against this concern:
> >
> > > I do not agree hat a spectral analysis is out of scope for this paper. In practice, w0 is the most important hyperparameter to vary training dynamics and the properties of the reconstructions. I do not see how models of SIREN training dynamics can ignore the choice of w0, especially as the experiments also empirically show how varying it changes the other identified order parameters.
> >
> > Firstly, we emphasize once again that we are studying properties of the NTK, not the reconstructions.  Secondly, we did not ignore $\omega_0$.  We have simply opted to empirically study the impact of this parameter, as opposed to analytically.  The overlap analysis in Section 4.1 is conducted over an experimental sweep that alters the value of $\omega_0$, and Section 4.2 contains an ablation study to observe the qualitative impact of $\omega_0$ on the order parameter trajectories.
> >
> > Secondly, we have expanded our empirical studies of the impact of $\omega_0$ on the order parameters.  Section F.1 considers its impact on the correlation lengthscale $\xi_{corr}$ and the asymptotic value of the $C_{NTK}$.  Furthermore, Section C.2 visually demonstrates the variation of the principal eigenvector of the NTK with increasing $\omega$.  One takeaway is that increasing $\omega_0$ increases the model's ability to memorize noise.
> >
> > We agree that a dedicated, theoretical analysis of $\omega_0$ would make for an interesting subsequent work, but as the paper currently stands, there is already sufficient novel contribution without this.

---

> > > ### Author Response · Authors · 2024-11-27
> > >
> > > Regarding the other concerns raised:
> > >
> > > > As in Appendix C, we see that many of the purported phase transitions do not actually occur a substantial number of times, and whether or not they do occur depends on the image itself.
> > >
> > > We expect some variance given the complexity of the dynamics.  Ultimately, initial conditions and hyperparameters have a huge impact on the behaviour of the system, which affects the strength/appearance of different phases (for example, in [Tangent Kernel Collapse](https://arxiv.org/pdf/2305.16427)).  This does not invalidate our study of the phenomenon when it does occur.  In particular, NTK alignment (as measured by $\text{AUC}(v_0, \nabla I)$) occurs in over 61% of the cases studied.
> > >
> > > This brings up another important point.  $\text{AUC}(v_0, \nabla I)$ is just one of the order parameters we track to detect NTK alignment.  As described in Section 4.1, we also consider evolution of $\text{CKA}(K_Y, K_{NTK})$ and the principal eigenvalue, both of which have been related to NTK alignment in the literature.  They are more sensitive to general shifts, as opposed to the appearance of edges specifically.  In particular, they are better behaved under the strategy we employed for change-point detection.  We discarded no runs when considering overlap statistics involving these measures of NTK alignment (as described in footnote 3, we only discard runs if one of the order parameters was unreliable).
> > >
> > > Thus, Appendix C is merely looking at the statistics for the runs we had to discard for certain order parameters.  We recognize we may have caused some confusion in our discussion in Appendix C, so have tweaked that.
> > >
> > > > Is the main takeaway that SIRENs struggle to fit images with sharp spatial gradients?
> > >
> > > Yes, and specifically, this difficulty is reflected in the accumulation of the gradient magnitudes $||\nabla_{\theta} f||$, which in turn leads to their prominence in principal eigenvector $v_0$ of the NTK.  We mention this point at the end of Section 3.3.
> > >
> > > Another way to understand this (described in Section C.1) is that, because edges are difficult to model, SIRENs learn to memorize their distribution.  Per (Shwartz-Ziv & Tishby, 2017), memorization occurs at the transition between the fast and slow phases of learning, explaining why edges appear in $v_0$ at this time.
> > >
> > > > Potential test set leakage
> > >
> > > We adopt a more standard paradigm to the super-resolution task of resizing the training image (which naturally includes an average of the high-resolution pixels) and using this to predict the underlying high-resolution image. The approach you link to is more like super-interpolation, as the point spread function of the pixel is kept at the original high-resolution, but the sampling rate is modified. To avoid any "test set leakage" we remove points that are in the same location on the two grids from the test set.

---

> > > > ### Author Response · Authors · 2024-11-27
> > > >
> > > > Just as an update, we have modified the main body of the paper based on your previous feedback (ie regarding shortening certain sections to make more space for additional results).  Likewise, we have updated our introduction to emphasize some of the points that have come up in our discussion.

---

> > > > > ### Comment · Reviewer_r32t · 2024-12-02
> > > > >
> > > > > Thanks again for the detailed response. I do appreciate that this analysis provides some rationale for why SIREN networks struggle to fit images with sharp spatial gradients (alongside point 3 below) and I am raising my score accordingly. I encourage future revisions to emphasize this more strongly in the writing.
> > > > >
> > > > > Broadly, IMO, the presented analysis of SIREN training dynamics is incomplete.
> > > > >
> > > > > > **1. Regarding whether the proposed mechanism does occur**: "_This does not invalidate our study of the phenomenon when it does occur._"
> > > > >
> > > > > I do not agree that the analysis can then explain SIREN training dynamics in general if the phenomenon it seeks to understand occurs in only a subset of training runs. Given 15 relatively simple foreground-based images to demonstrate this phenomenon, this is insufficient for analyses that are based on a significant empirical component.
> > > > >
> > > > > > **2. Regarding understanding w_0**: "_We did not ignore w_0. We have simply opted to empirically study the impact of this parameter, as opposed to analytically._"
> > > > >
> > > > > I understand this, but the claim is to provide a theoretical NTK alignment-based framework to understand SIREN training dynamics (to quote the rebuttal: "_our contributions are primarily theoretical_"). If the presented framework does not account for the most important hyperparameter for empirical SIREN training, I do not think that it is complete.
> > > > >
> > > > > > **3. Regarding finding edges in the reconstruction vs principal eigenvector**
> > > > >
> > > > > My apologies for the typo and confusion, I had the principal eigenvector in mind with that comment (responded to [here](https://openreview.net/forum?id=muN3B40keb&noteId=UjeaimNCtc)) instead of the reconstruction. I do think that the rebuttal to that point made a good argument regardless for why it is worth investigating and not obvious.

---

> > > > > > ### Author Response · Authors · 2024-12-03
> > > > > >
> > > > > > Thank you for all of your responses throughout the discussion period.  We are very grateful for the higher score.  Though this discussion period is coming to an end, we would still like to challenge some of your remaining remarks.
> > > > > >
> > > > > > > ...the phenomenon it seeks to understand occurs in only a subset of training runs.
> > > > > >
> > > > > > We appreciate the reviewer's concern regarding the generality of our analysis. We acknowledge that certain individual measures, such as $AUC(v_0, \nabla I)$, may not fully capture the nuances of the phenomenon in all training runs. For instance, Figure 12 in the supplementary material illustrates cases where visual inspection clearly reveals edges in the principal eigenvector, even when the $AUC(v_0, \nabla I)$ value is less than 0.6. This highlights the limitation of relying solely on the AUC as a definitive indicator of edge alignment. The AUC, while useful, primarily measures the _sharpness_ of this alignment. A lower AUC value doesn't necessarily indicate the absence of the phenomenon, but rather a less distinct alignment.
> > > > > >
> > > > > > To this end, we also track other order parameters such as CKA with the target kernel ($K_Y$) and the divergence of the principal eigenvalue, which capture different facets of NTK alignment. These alternative measures consistently exhibit a sudden change at the phase transition, _even in cases where the AUC is low_. This convergence of evidence from diverse measures, capturing different aspects of the phenomenon, strengthens our claim that this is a general characteristic of SIREN training dynamics, rather than an artefact of specific runs or images.
> > > > > >
> > > > > > Similarly, for the loss rate collapse, while zero-crossings of $\min C_{NTK}$ or peaks in $\sigma_{\theta}$ might not be detectable in every run, the inflection points in the evaluation loss ($L_{eval}$) consistently indicate the transition from "fast" to "slow" learning (Figure 4, right). The clustering of these change points across runs, as observed in Figure 4, further supports the generality of the phenomenon. The varying sensitivities of these measures allow us to robustly identify the phase transition even when individual metrics might be less informative in specific scenarios. Therefore, the combined evidence from these multiple, independent measures provides a stronger basis for our analysis of SIREN training dynamics.
> > > > > >
> > > > > > > Regarding understanding w_0
> > > > > >
> > > > > > We appreciate the reviewer's point regarding the importance of $\omega_0$ in SIREN training. While our analysis doesn't explicitly model $\omega_0$, it does identify a key effect – the control of the NTK's diagonal dominance and consequently the correlation lengthscale. This connection is crucial, and is supported by existing work such as [link](https://openreview.net/forum?id=yVqC6gCNf4d) and [link](https://arxiv.org/abs/2112.01917), which directly demonstrate how $\omega_0$ governs the NTK's diagonal dominance. These findings provide a strong foundation for the locality assumption central to our analysis, an assumption we further validate experimentally (Figure 16).
> > > > > >
> > > > > > That said, our focus is on the dynamically evolving correlation lengthscale $\xi_{corr}(t)$, rather than the static initialization controlled by $\omega_0$.  This is motivated by its greater relevance to the _spatial aspects_ of the training dynamics we study (ie spatial edge alignment, rather than frequency alignment). While $\omega_0$ influences the initial $\xi_{corr}$ (and to some extent its asymptotic value, as shown in Figure 16), it's the time-dependent behavior of $\xi_{corr}(t)$ in the spatial domain that directly impacts edge alignment.  We analyse this in Sections 3.1 and 3.2, where we tie it to the spatial variations in the weight gradients $\nabla_{\theta}f$.
> > > > > >
> > > > > > We agree that further investigation into the role of $\omega_0$ could be fruitful. For example, refining Equation (26) to explicitly incorporate $\omega_0$ and its influence on edge selection (as observed in the principal eigenvector noise) is a promising direction for future work. However, we believe that our current theoretical contributions, including Equation (26), provide a sufficient framework for understanding the core aspects of SIREN training dynamics, particularly the NTK alignment phenomenon and its relationship to the fast-to-slow learning transition. By leveraging the established understanding of $\omega_0$'s impact on the NTK, as detailed in the cited works, we believe our framework effectively captures the essential influence of $\omega_0$ without requiring explicit modeling within our equations.

---

### Official Review · Reviewer_LUop · 2024-11-03

**Soundness:** 4
**Presentation:** 3
**Contribution:** 4
**Rating:** 6
**Confidence:** 4

**Summary:**

This is a quite interesting paper that considers the training dynamics
of sinusoidal neural networks. It takes an approach that is in the
tradition of understanding machine learning methods through a
"statistical physics" approach, seeking to understand
disordered/ordered phenomena through phase transitions.

Specifically, this work concerns itself with examining the dynamics of
training sinusoidal networks fit to image data via the evolution of
the neural tangent kernel. That is to say, by examining how the
correlation structures of the gradients evolve, the patterns of
learning dynamics are uncovered. Multiple methods and metrics are
proposed to detect these changes, which are shown to correspond to
distinct phases and phenomena in fitting sinusoidal networks to image
data. These metrics seem to provide a nice mix of "local" and "global"
measurement.

**Strengths:**

This paper provides a nice way to examine the behavior of neural
networks used for representing images by examining the features of
tangent kernels. I found it quite easy to follow, and reasonable in
its claims. For the most part, the authors justified their ideas
without making overly-bold claims regarding their derivations -- in
this way, the work is modest, which is appreciated.

I also found the metrics proposed to be practical -- the quantities
studied are fairly easy to calculate, which suggests that this paper
could be of use to practitioners.

Although some of the claims aren't entirely justified (for instance,
see the discussion on "diffusion" in the weaknesses section), for the
most part the metrics proposed by the authors for detecting do not
come out of thin air -- they are justified by a mixture of approximate
calculations and empirical insights gleaned from tests on image data.

**Weaknesses:**

Before listing the weakness: I would like the authors to understand
that the list of weaknesses is much longer than the list of strengths
not because it is a bad paper, but because the weaknesses are
"actionable," and thus are better-served by a more detailed write-up!

It is unclear to what extent the results are specific to sinusoidal
networks. Of course, images parameterized by SIREN networks are used
to demonstrate the ideas, but I as the reader am not sure whether the
analysis carried out in the paper is really specific to anything about
natural images, apart from some references to the notion of an "edge"
in an image in Section 3.2. It would be helpful if the authors would
clarify what aspects of their work they think are specific to
images/SIREN, and what aspects should apply broadly to regression
tasks using MLPs. A simple question to ask is the following: does it
make sense for this paper to be rewritten using ReLU MLPs applied to
low-dimensional regression tasks (which I suppose would look more like
the references [23,24,25])? In this direction, a more detailed
comparison to the existing literature would be helpful -- as it is
now, your proposed metrics are not compared to anything else in the
literature.

The arguments made by the authors to characterize, say, the phenomenon
of "diffusion" (Eq. 15), are a bit handwavy, in the sense that they
make a series of approximations that end up in a nice PDE that one can
call a diffusion equation. It is plausible that this is the underlying
mechanism explaining the diffusion phase in the training dynamics, but
it is not clear if this is a coincidence or not. Some more validation
in the experiments section would have been nice, showing somehow that
the training dynamics indeed resemble a diffusion equation like the
one given in (Eq. 15). For example, if we were to run two experiments:
one where we perform gradient descent to fit an INR to an image, and
another where we use the gradients of the INR to parameterize the
diffusion equation (15), would they yield similar results? Something
along these lines would make the claim significantly more convincing.

Despite the analysis being interesting, the conclusions fell flat at
the end. I understand that this paper is more of a scientific
investigation than it is an engineering exercise, so I do not fault
this work for not providing extensive experiments, proposed methods,
and so on. However, the audience of this work will probably go beyond
those who are interested in inherent properties of INRs -- the only
section to this effect was Section 4.2, which had the single
conclusion saying that $\omega_0$ is the most important
hyperparameter, as opposed to width and depth, say. It would be nice,
either via more experiments or an expanded discussion, to see how the
proposed metrics could be used by practitioners to guide their design
and use of INRs.

Another /potential/ weakness of this paper is the small scale of the
experiments. This doesn't bother me too much, personally, but it would
be more convincing if your results were reported for a larger set of
images.

**Questions:**

Please see the weaknesses section for points that would be helpful to have addressed -- I think addressing those points thoroughly would strengthen this paper by quite a bit.

I also have some editorial notes:

The references in this paper are formatted in a non-standard
way. Please make sure that the paper formatting guidelines are
followed closely. I am not sure if the way the authors have formatted
their in-text references is wrong, but it is different than most ICLR
papers I have read.

There are also some formatting issues: e.g., at the beginning of
Section 3.1 "equation equation 5." Please look over the manuscript
carefully to check for these sorts of errors.

Figures repeatedly are missing useful features such as colorbars,
clear axis labels, and so on. Please look over your figures and work
on their clarity!

---

> ### Author Response · Authors · 2024-11-20
>
> We are very grateful for the detailed review. We have the following comments:
>
> > It is unclear to what extent the results are specific to sinusoidal networks...A simple question to ask is the following: does it make sense for this paper to be rewritten using ReLU MLPs applied to low-dimensional regression tasks (which I suppose would look more like the references [23,24,25])?
>
> We have added a new section to the supplementary materials (section D) which compares SIREN models to ReLU-MLPs with a positional encoder, as used in [44].  Our main empirical takeaway is as follows: while ReLU-PE models also exhibit Neural Tangent Kernel alignment, it is a much slower, non-local process, that does not coincide with loss-rate collapse or translational symmetry breaking.  In contrast, in SIRENs, the NTK is dominated by its local structure, and NTK alignment takes place alongside loss rate collapse, translational symmetry breaking, and the onset of diffusion.
>
> > It is plausible that this is the underlying mechanism explaining the diffusion phase in the training dynamics, but it is not clear if this is a coincidence or not...{if} we use the gradients of the INR to parameterize the diffusion equation (15), would they yield similar results?
>
> Thank you for the feedback.  We think that the diffusion behaviour is a consequence of the locality of the NTK, as seen in Figure 2.  Intuitively, due to locality, a residual at a point will only change significantly if it is has neighbours with high residuals.  Over time, this disturbance propagates to the neighbours' neighbours, resulting in the formation of a propagating wavecrest. Our derivation of the diffusion equation merely expresses this insight mathematically.
>
> Regarding the comparison between the solution of the diffusion equation and the true gradient dynamics, this is something we have thought a lot about.  While using the gradients of the INR might be a tricky, one nice thing about our diffusion equation is that it is very easy to identify the steady-state solution:
>
> $$
> \Delta^2 r(x) = -\frac{2}{\xi^2(x)} r(x)
> $$
>
> This is just a Helmholtz equation, which would be possible to analyze using numerical methods.  However, defining "similarity" in this context - that is, comparing the analytical solution to that obtained by gradient descent - requires some careful thinking.  We would be very open to hear suggestions.
>
> That said, we think this analysis may fall outside the current scope of the paper, which is more concerned with deriving, and understanding the relation between, different order parameters that summarize the phases of learning.  In deriving the diffusion equation, our goal was to identify the order parameters that described the onset of the wave behsviour (which we found were the correlation lengthscale, and the asymptotic value of the $C_{NTK}$).  The remainder of the paper then examines how the evolution of these parameters matches the evolution of other order parameters.
>
> > It would be nice, either via more experiments or an expanded discussion, to see how the proposed metrics could be used by practitioners to guide their design and use of INRs.
>
> Our work reinforces previous observations about the critical roles played by gradient confusion [13] and correlation length-scale [12] in determining the dynamics, and generalization behaviour, of DNNs.  Though we do not claim these as a contribution, in our paper, for SIREN models specifically, the locality of the NTK has allowed us to derive alternative ways to calculate proxies for these metrics, which are potentially easier to compute, and thus guide practitioners. The utility of metrics like MagMa and $AUC(v_0, \nabla I)$ towards this end would be interesting as future work, but at present we feel it is outside the scope of the paper.

---

> > ### Comment · Reviewer_LUop · 2024-11-26
> >
> > Thank you for your response -- I think the added appendix on the differences with ReLU networks is a nice addition. I am not inclined to change my score at this point, given the somewhat limited theoretical contributions apart from empirical observation; the merely observational comparison to diffusion equations is still troublesome. I do appreciate the scientific approach to studying these methods, but the paper overall does not warrant a strong response from me.

---

> > > ### Author Response · Authors · 2024-11-28
> > >
> > > Thank you for your response.  We respectfully believe the theoretical contributions of this work are more substantial than currently perceived, and we would like to clarify the significance of our findings.  We agree that some aspects rely on empirical validation, but these observations are grounded in and supported by the theoretical analysis.  Specifically:
> > >
> > > 1. **Derivation of the local approximation of the Cosine NTK:** Our core theoretical contribution lies in deriving the local approximation of the Cosine NTK in Section 3.2, based on a Cauchy Distribution. This derivation is not merely observational; it provides a rigorous analytical foundation for understanding the behaviour of SIRENs.
> > > 2. **Explicit formulas derived from the approximation:** This approximation allows us to derive explicit, closed-form formulas for crucial properties: (a) the correlation lengthscale, (b) the minimum of the Cosine NTK, and (c) the principal eigenvector of the NTK. These formulas are not simply observed; they are derived analytically and offer valuable insights into the dynamics of SIRENs.
> > >
> > > To our knowledge, this work represents the first study of dynamic NTKs in SIRENs, including the derivation of the aforementioned quantities. Furthermore, within the broader context of NTK alignment, we believe our analysis is novel in performing such a detailed theoretical characterization for a deep, nonlinear network. We are unaware of prior work achieving this level of analytical depth for similar architectures. If you know of any such work, we would be grateful for the references, as they would be valuable for contextualizing our contributions.

---

### Official Review · Reviewer_EyqR · 2024-11-04

**Soundness:** 3
**Presentation:** 3
**Contribution:** 3
**Rating:** 6
**Confidence:** 3

**Summary:**

The authors report interesting theoretical work and computational experiments to analyze transitions in the evolution of parameters of a neural network. They use SIREN (sinusoidal representation networks) models of simple natural images (cameraman, cat) for computational experiments. They unify several order parameters, namely the evolution of loss distribution, the evolution of loss rate, and the evolution of the prediction over a local neighborhood of data points to identify the fast and slow training phases. An important feature of the work is the interpretation of the spatial order of neural tangent kernels during different phases of training.

**Strengths:**

* The analysis of training dynamics in SIRENs is sound. Understanding training dynamics in neural networks is important as the field works towards foundational and generalist models. The authors establish unifying links between different measures of phase transitions in a neural network's parameters.
* The order parameter (MAG-Ma) is a valuable measure of spatial alignment in the model's predictions. The authors also introduce an order parameter that measures spatial alignment with the data's spatial structure.

**Weaknesses:**

* Writing: The description is dense in critical parts of the paper and not always sequential. For example, the transition from eq. 2 to eq. 3 requires an elaboration: the time derivative of parameters ($\dot{\theta}$) is the same as the opposite of the gradient of the loss relative to the parameters. After articulating this conceptual link, eq. 3 can be derived from eq. 2. Before that, eq. 3 is too dense.
* No links to CNN: Given the prevalence of CNNs in discriminative and generative models of images, it is important to contrast phase transitions in CNNs vs INRs. Without the comparative analysis, it appears that MAG-Ma can be used only for INRs and not other models that depend on the edges/spatial texture for regression.
* Computational experiments are insufficient: The authors should assess the training dynamics for a substantive dataset, e.g., each image of ImageNette.
* No code seems available: Reproducing and building on top of this work requires authors to share their code. Some of these order parameters can be used to improve training efficiency, e.g., to stop training once the critical point is reached.
* I'd suggest an alternative title to clarify the contribution (as I read): "Surfacing inductive biases of sinusoidal networks by linking phase transitions”

**Questions:**

* Are the data in Fig. 2, 5, and 6 that show the mean and variance of the order parameters measured over multiple training runs for the same image (cameraman or cat) or different images?  If single images, why not evaluate training dynamics over a substantive set, e.g., a set of models for 1000 different images that represent natural images and physical simulations?
* Can the order parameters be computed efficiently during training to be used as stopping criteria or to adjust the learning rate?
* An observation: the neural tangent kernel is mathematically related to the concept of spatial coherence (measured by mutual intensity) in statistical optics (see Joseph Goodman's book on statistical optics).

---

> ### Author Response · Authors · 2024-11-20
>
> We are very grateful for the detailed review.  We have the following comments:
>
> > It is important to contrast phase transitions in CNNs vs INRs
>
> Thank you for your feedback.  It is crucial to note that training Implicit Neural Representations (INRs) and Convolutional Neural Networks (CNNs) are fundamentally different learning problems with distinct characteristics. In this case, we train an INR to represent a **single** image, where the inputs are pixel coordinates, and the outputs are grayscale values.  The resulting model is a learned continuous representation of a single image.
>
> By contrast, CNNs are trained on **multiple** image to create predictors for a task that generalise to new data.  In an INR, each pixel coordinate/grayscale value is a distinct training datapoint, whereas in a CNN, each pixel grayscale value is a distinct input feature. Thus the impact of "edges" are very different in both models.  In CNNs, models are encouraged to learn filters which are templates of edges.  However, the relationship between edges and model weights is much more subtle in INRs. Our paper illustrates that insight can be gained from the phenomenon of Neural Tangent Kernel alignment.  For example, we demonstrate that the principal eigenvector for the NTK of a SIREN is sensitive to data points where the gradient $\nabla_x f$ is high.  In the case of a CNN classifier, the same eigenvectors would not detect edges in images, but rather, edges between image manifolds (e.g. an image that is midway between being a cat and a dog).
>
> Given how different the use cases, the architectures, and the conclusions are for INRs and CNNs, we don't think it is reasonable to consider CNNs, and their phase transitions, in the main body of the text.
>
> > Computational experiments are insufficient
>
> Our revised submission contains results from ten additional images (full set shown in Figure 9), which we do draw from ten different ImageNet classes.  We believe performing our experimental sweep on each image of ImageNet (as you recommend) is impractical - with our variations in seeds, architecture, and hyperparams, this would necessitate us to train on the order of 100M models.
>
> We do recognize that some interesting results could be found from such a study.  From the preliminary analysis in Section C, we note that NTK alignment and loss rate collapse do not always take place, and there is some relationship between the occurrence of these phase transitions and the complexity of the image.  While tackling this would certainly make for a thought-provoking paper, we think it is out of scope for the present work, but certainly worth looking at in the future!
>
> > Why not evaluate training dynamics over a substantive set, e.g., a set of models for 1000 different images
>
> We have included more figures in Appendix G.  As outlined above, each image produces one INR, so the order parameter trajectories can only be collected for one image at a time.  The specific timings for the evolution of the order parameters vary immensely across different architectural choices and images.  In Figure 11 of the Appendix, you can also see all runs plotted on a single histogram, comparing SIRENs with ReLU-Pe.
>
> That said, the high variance of the order parameters between runs makes these sorts of plots very cluttered.  Instead, we compare the relative timings between critical points of the order parameter trajectories (for example, looking at how certain critical points tend to cluster).  An example is the right side of Figure 4, which examines the co-occurence of detected phase transitions, aggregated across all architectures and images studied.
>
> > Can the order parameters be computed efficiently during training to be used as stopping criteria or to adjust the learning rate?
>
> It is important to emphasize that many of the order parameters considered in this work are themselves referenced from other works that investigated just this question.  Indeed, our work reinforces previous observations about the critical roles played by gradient confusion [13] and correlation lengethscale [12] in determining the dynamics, and generalization behaviour, of DNNs.
>
> For SIREN models specifically, the locality of the NTK has allowed us to derive alternative ways to calculate proxies for these metrics.  In particular, our decomposition of the NTK (A.1) and the local structure (A.3.2) have a lower memory footprint than direct computation.  However, we are dubious about extending this approximation to higher-dimensional datasets.
>
> MAG-Ma is efficient to compute, and would be interesting to explore in future work.  Presently, we feel it's outside the scope of the paper.
>
> > see Joseph Goodman's book
>
> Thank you for pointing us towards this reference!  We have added a citation to the Goodman book in the main text.

---

> > ### Comment · Reviewer_EyqR · 2024-12-03
> > **review of the rebuttal**
> >
> > I thank the authors for a thorough revision of their work and detailed responses. The revised paper better articulates the scope and utility of the order parameters for understanding the training dynamics of INR. I have revised my rating for presentation and recommendation for acceptance.

---

### Official Review · Reviewer_UZMn · 2024-11-04

**Soundness:** 3
**Presentation:** 3
**Contribution:** 3
**Rating:** 6
**Confidence:** 1

**Summary:**

The paper identifies a shared, underlying mechanism connecting three seemly distinct phase transitions in the training of a class of deep regression models, in implicit neural representations of image data.  The paper then presents experimental results on the distribution of critical points and the dynamical consequences of hyperparameters.

**Strengths:**

The paper is easy to follow.
The paper conducts 'preliminary investigations into the dynamics of feature learning within INRs for image data' and analyzes the dynamic phase transition of SIREN models.  This shows 'NTK provides a rich theoretical tool for deriving and relating order parameters to understand training dynamics'

**Weaknesses:**

I am unfamiliar with this field.
As the paper is mainly theoretical, it is unclear beyond the simple raw images presented in the paper. In my opinion, such theoretical works tend to be less useful than application papers in general since they only tend to be valid in very specific circumstances.   Furthermore, there are no practical suggestions on improving machine learning performance.  However, as there are other theoretical papers that analyze NTK that are accepted to peer machine learning conferences this paper is within the remit of ICLR.

**Questions:**

N/A

---

> ### Author Response · Authors · 2024-11-20
>
> Thank you for taking the time to read our paper.   We appreciate your feedback and understand your perspective on theoretical work. While the work is presently specialized, we do believe it contains important observations about learning in a more complex set of models than is usually studied in the NTK alignment literature, and as such represents a step towards practical advances in the field.

---

> ### Comment · Reviewer_UZMn · 2024-11-21
> **Response to the Authors**
>
> Thank you for the reply.
>
> > it contains important observations about learning in a more complex set of models than is usually studied in the NTK alignment literature, and as such represents a step towards practical advances in the field.
>
> Could you elaborate on this point in detail?

---

> ### Author Response · Authors · 2024-11-21
>
> Happily.  On page 10, in the related works section, we describe some of the models that are commonly studied in the NTK alignment literature.  A common strategy is to analytically derive insights from linear models or deep linear models where the evolution of the NTK is more tractable, and then empirically show that the insights translate to nonlinear architectures [23,24,35].  By contrast, the models we study, SIRENs, are fully nonlinear (sinusoidal activations), deep (3-7 layer) neural networks.  What makes our analysis tractable is that SIRENs are trained on low dimensional inputs, in this case, single 2D images.  Thus, it is much easier to identify how the individual datapoints, and the image gradient at those points, impact model learning.  In particular, we focus on why the principal eigenvectors of the NTK align with these image gradients during training.

---

### Author Response · Authors · 2024-11-20

We would like to thank all the reviewers for their detailed responses.  We look forward to engaging in discussion.  Based on the feedback, we would like to highlight the following changes in our revision:

- We include more experimental results in the paper.  Our revised submission contains results from ten additional images (15 total, full set shown in Figure 9).  In addition, we include benchmark comparisons with ReLU-PE models in Section D of the supplementary materials.  In addition, we include more numerical experiments to gauge the fidelity of our experiments (building upon the previous section D, now section F), and more visualizations of the runs (similar to Figures 4-6) in Section G of the supplementary materials.  These results all reinforce our previous claims, but now provided additional rigor.
- We have made minor modifications to the change detection algorithms (described in B.2), and the aggregation strategy for examining the coincidence of phase transitions across runs.  In particular, we include a more detailed investigation of the failures of the change detection algorithms, which we tie to properties of the images in Section C.
- We have made more explicit references to the Supplementary materials throughout the main body of the paper, to make it easier to find supporting arguments and experiments.  We have also cleaned up some minor elements of the presentation.  In particular, we emphasize that that the validation set we use to measure model performance is a super-resolution task.

---

### Author Response · Authors · 2024-11-27

Per ongoing discussion, our new revision includes the following modifications:

- We have re-organized our introduction in order to more clearly communicate the theoretical goals of our work - namely, to extend the existing NTK Alignment literature to consider more sophisticated models, such as SIRENs.
- We have included more empirical analysis of the impact of $\omega_0$ on NTK alignment and the local structure of the NTK in Appendix C.2 and F.1, which we link to in Section 3.1 and 4.1.  In particular, we clarify a previous miscommunication regarding the occurrence of NTK alignment.
- The references in the paper are now correctly formatted.

---

### Author Response · Authors · 2024-11-27

Per ongoing discussion, in an effort to improve our presentation and better clarify our contributions, we have made the following updates to the paper:

- We have modified the description of our contributions at the end of the Introduction, to better emphasize the numerous theoretical contributions contained in our paper, and the takeaway messages.
- We have similarly updated the conclusion to better tie together the insights derived in the paper.
- We have shortened some sections in the main body of the paper, and swapped some figures, in order to make space for the expanded set of result that were previously only in the Supplementary Materials (once again, to make our contributions more apparent).

---

### Meta-Review · Area_Chair_mJ6Z · 2024-12-20

**Metareview:**

This paper studies the training dynamics and feature learning in SIRENs---implicit neural networks with sinusoidal activations. The authors use the formalism of neural tangent kernels (NTK) and NTK alignment to characterize three phases of learning in SIRENs. While there was an agreement among the reviewers that the paper contains interesting material, there was similarly a consensus that the current version is at best a borderline case.

Here is a list of things that could be improved:

- The authors rightly point out that they analyze NTKA in a real nonlinear DNN, but prior work on simplified models seeks rigorous results (either math or physics style). While this is indeed a physics-style paper, some derivations still be done more carefully and approximations should be better justified
- Several reviewers asked whether any of the results can be mapped to other implicit neural nets; the authors commented on this but it would be good to have a more substantial discusssion which elaborates, perhaps theoretically and numerically, on the different behavior of SIREN- and NeRF- style networks already in the main text. This is clearly interesting.
- r32t pointed out that the experiments are far too preliminary for clear takeaways; indeed the original version contained experiments on only 5 images and the revision on (still only) 15. In line with a question from LUop---how is any of this specific to natural images---it would be interesting to see what happens for images that don't look anything like natural images (e.g. Fourier transforms of natural images, modulation images, etc, ...)
- run still more convincing, larger-scale experiments that unambiguously support the theoretical inferences proving that the observed phenomenology is robust over image classes, resolutions, network sizes and hyperparameters, but also delineate regions where it doesn't happen
- further discussion, even if speculative, on causal mechanisms behing the observed phenomenology (but also on cases where it is absent)

At the moment this is a paper which is slightly below the acceptance quality. I am convinced that with these points addressed the paper will do very fine.

**Additional Comments On Reviewer Discussion:**

The discussion was respectful and productive. UZMn asked whether there is a way to operationalize the findings to improving training, learning rate scheduling, etc. This is an important question and something for the authors to consider. It would've been nice to see at least a speculation in the response. EyqR praised the introduced MAG-Ma metric but criticized clarity, experiments, lack of code, and title. The authors' response pleased EyqR and they bumped their rating. LUop acknowledged the interesting observations but criticized scope (SIRENs vs all INRs; natural vs any images, ...), handwavy approximations. The authors introduced supplementary material on NeRFs but LUop did not feel it's sufficient to change score. r32t criticized scope and rigor of experiments and presentational issues. There was a rich ensuing discussion but r32t remained on the mildly negative side. s3fX made precise technical remarks which proved to be real problems; the authors fixed them and s3fX bumped their score, though reluctantly ("although I am not enthusiastic...").

---

### Decision · Program_Chairs · 2025-01-22

Reject